# The Role of Prostate Combination Biopsy Consisting of Targeted and Additional Systematic Biopsy

**DOI:** 10.3390/jcm10214804

**Published:** 2021-10-20

**Authors:** Chung Un Lee, Joongwon Choi, Si Hyun Sung, Jae Hoon Chung, Wan Song, Minyong Kang, Hyun Hwan Sung, Byong Chang Jeong, Seong Il Seo, Seong Soo Jeon, Hyun Moo Lee, Hwang Gyun Jeon

**Affiliations:** 1Samsung Medical Center, Department of Urology, School of Medicine Sungkyunkwan University, Seoul 06351, Korea; iatronices@naver.com (C.U.L.); ssh.sung@samsung.com (S.H.S.); dr.jhchung@gmail.com (J.H.C.); wan.song@samsung.com (W.S.); dr.minyong.kang@gmail.com (M.K.); hyunhwan.sung@samsung.com (H.H.S.); bc2.jung@samsung.com (B.C.J.); seongil.seo@samsung.com (S.I.S.); seongsoo.jeon@samsung.com (S.S.J.); hyunmoo.lee@samsung.com (H.M.L.); 2Department of Urology, VHS Medical Center, Seoul 05368, Korea; uromedic@outlook.com

**Keywords:** prostate biopsy, Prostate Imaging Reporting and Data System (PI-RADS), combination, targeted biopsy, systematic biopsy

## Abstract

Background: To identify the role of combination biopsy, which consists of both targeted and additional systematic cores, in the diagnosis of clinically significant prostate cancer (csPCa). Methods: We retrospectively reviewed patients with PSA levels 2.5–15 ng/mL who have a suspicious prostate lesion (with the Prostate Imaging Reporting and Data System (PI-RADS) ≥ 3) on multiparametric MRI (mpMRI) between January 2016 and December 2018. We analyzed biopsy results by PI-RADS score and biopsy methods (systematic, targeted, and combination biopsy). Results: Of the 711 total patients, an average of 4.0 ± 1.8 targeted and 8.6 ± 3.1 additional systematic biopsies were performed. The additional systematic biopsies were sampled outside the targeted biopsy area. The combination biopsies detected more csPCa (201 patients, 28.3%) than did the targeted (175 patients, 24.6%) or systematic (124 patients, 17.4%) biopsies alone (*p* < 0.001). In the initial biopsy samples, there was a 7% increase in the detection of csPCa than in targeted biopsy (62% to 69%). It increased by 11% in repeat biopsy (46% to 57%). There was no statistical significance in both groups (*p* = 0.3174). Conclusions: Combination biopsy has the benefit of detecting csPCa in both initial and repeat biopsy when there is a suspicious lesion on mpMRI.

## 1. Introduction

Prostate cancer is the most common cancer among American men and has accounted for 20% of newly developed male cancers in 2019 [1]. It is the fourth most common cancer among men in Korea [2]. Currently, the most common screening method for prostate cancer is the prostate-specific antigen (PSA) [3], and PSA has been one of the criteria for performing a prostate biopsy.

The decision to perform a prostate biopsy is typically made based on not only PSA value but also when there is a hypoechoic lesion on the ultrasound or a suspected lesion on the MRI or digital rectal exam (DRE) findings. However, once a biopsy is planned, there remains debate regarding which biopsy method is best for cancer detection. Since Hodge introduced the sextant biopsy, various methods of prostate biopsy have been studied [4], such as transrectal systematic biopsy, multiparametric MRI (mpMRI)-based targeted biopsy, combination biopsy, and transperineal saturation biopsy.

The advantages of the Prostate Imaging Reporting and Data System (PI-RADS) in prostate cancer diagnosis have been described. Version 2 of this system is also known to be more beneficial than the first version [5,6]. MRI-targeted biopsy was based on this scoring method. There are reports that sextant biopsies miss clinically significant prostate cancer in up to 37% of cases [7]. Therefore, twelve core transrectal systemic biopsies are generally pursued by cancer detection. However, the advantages or specific indications for systematic biopsy, MRI-targeted biopsy, and combination biopsy remain unclear. The combination biopsy consists of systematic and MRI-fusion targeted biopsy. There is a study that combination biopsy has better diagnostic performance than other biopsy methods, but heterogenous reports were present [8]. Moreover, there have been few reports of targeted biopsy with additional systematic biopsy [9].

We aim to compare targeted and additional systematic biopsy in combination biopsy methods with regard to cancer detection or detection ratio of clinically significant prostate cancer (csPCa, G/S ≥ 7). We were also interested in determining which biopsy technique is better in PI-RADS 3, 4, or 5 diseases based on mpMRI findings. Finally, we compared each biopsy technique with initial and repeat biopsy patients.

## 2. Materials and Methods

### 2.1. Ethics Approval

This study was approved by the Institutional Review Board of Samsung Medical Center (IRB No. 2019-08-159), and the IRB waived the requirement for informed consent due to the retrospective nature of this study. All methods were carried out in accordance with the Declaration of Helsinki.

### 2.2. Study Population

A total of 2937 patients underwent prebiopsy mpMRI and subsequent biopsies at our hospital between January 2016 and December 2018. Seven hundred and eighty-two patients with PSA < 2.5 ng/mL or >15 ng/mL and 1441 who performed a systematic or targeted biopsy only were excluded. Three patients diagnosed with urothelial cancer were also excluded. Finally, 711 patients with a PSA of 2.5–15 ng/mL who had undergone combination biopsies (targeted with additional systematic) following the mpMRI with PI-RADS ≥ 3 lesions were included in the analysis.

### 2.3. Study Design

The PSA cut-off value is usually 3.0 or 4.0 ng/mL, but 2.5 was also considered significant for Koreans in previous studies, so it seems appropriate to use 2.5 as the standard. We used 2.5 ng/mL as a cut-off value because a prior study from our group demonstrated that patients in whom PSA values were 2.5–4.0 and 4.0–10.0 ng/mL had similar rates of prostate cancer and characteristics of prostatectomy specimen [10,11].

We investigated the highest Gleason score (G/S) by separating the targeted and systematic biopsy cores from the combination biopsy results and comparing any differences in the G/S distribution. In subgroup analysis, the G/S distribution was compared in the initial and repeat biopsy settings. We also compared the differences in the rates of csPCa (csPCa/total cancer core) according to each biopsy method (targeted, systematic, and combination) between the PI-RADS scores (3, 4, and 5).

### 2.4. mpMRI and Biopsy Protocol

All patients underwent mpMRI on a 3-Tesla MRI system with triplanar T2-weighted, diffusion-weighted, and dynamic contrast-enhanced sequences without an endorectal coil. mpMRI images were analyzed according to the Prostate Imaging Reporting and Data System version 2 (PI-RADS v2). In patients with a suspicious region of interest, defined as a PI-RADS score 3 or higher on mpMRI, targeted cores were obtained after fusion (25.8%) or cognitive biopsy (74.2%). The additional systematic biopsy was sampled outside the targeted biopsy area, and the number of cores was determined by radiologists who performed the biopsy. Two experienced radiologists decided on biopsy method in accordance with their usual practice. There are concerns about the risk of missing tumors of the anterior prostate and apex when performing a biopsy through the transrectal approach. However, there are studies that reported MRI-targeted TRUS biopsy can sample the anterior prostate and apex with significant torque [12,13]. In our study, experienced radiologists performed MRI-targeted TRUS biopsy, and the probability of missing biopsy of the suspicious region of interest was thought to be low.

### 2.5. Statistical Analysis

All results are presented as medians with interquartile ranges or numbers with percentages. We used the Kolmogorov–Smirnov test to analyze the continuous variables for normality. The Chi-square test was used to analyze the categorical variables. The independent *t*-test and Mann–Whitney U-test were used to analyze the descriptive variables according to normal distribution. Statistical analysis was performed using SPSS version 21.0 (IBM, Chicago, IL, USA) and R 3.5.1 (R Core Team, Auckland, New Zealand).

## 3. Results

Table 1 shows the baseline demographics. Of all patients, the PI-RADS score distributions of 3, 4, and 5 in mpMRI were 199 (28.0%), 403 (56.7%), and 109 (15.3%), respectively. Four hundred and eleven (57.9%) cases were initial biopsies, while 300 (42.1%) were repeat biopsies. An average of 4.0 ± 1.8 cores of targeted biopsy and 8.6 ± 3.1 cores of additional systematic biopsy was performed. In total, we gathered an average of 12.5 ± 2.1 cores.

Table 2 demonstrates the biopsy results. In PI-RADS 3, 4, and 5 groups, the cancer detection rate increased gradually from 16.6 to 47.1 to 80.7%, respectively (*p* < 0.001). There were also significant differences in the combination, targeted, and systematic core G/S between the three PI-RADS groups. In contrast, targeted core upgrading had no significant correlation with the PI-RADS score (*p* = 0.651).

Combining the targeted and systematic biopsies resulted in a higher detection rate of csPCa (201 patients, 28.3%) than did targeted (175 patients, 24.6%) or systematic (124 patients, 17.4%) biopsy methods alone (*p* < 0.001, Figure 1). In the initial biopsy samples, there was a 7% increase in the detection of csPCa than in targeted biopsy (62% to 69%). It increased by 11% in repeat biopsy (46% to 57%). There was no difference in the diagnosis rate change in both groups (*p* = 0.3174, Figure 2). Among 201 patients diagnosed with prostate cancer, only 55 patients (27.4%) were diagnosed with csPCa based on systematic biopsy alone. Twelve additional patients were identified to have csPCa with concomitant targeted biopsies. On repeat biopsy, the additional number of cores identified 53.7% more csPCa (63/41) on combination biopsy when compared to systematic biopsy. Among 110 patients diagnosed with cancer, 22 were detected to have csPCa using systematic biopsies alone. The csPCa rate (csPCa/total cancer core) in combination biopsies (in the PI-RADS 3, 4, and 5 groups) was 7–10% higher than it was with targeted biopsy alone in all three groups (*p* < 0.001, Figure 3). After additional systematic biopsy, a portion of insignificant cancer increased from 12.7% to 15.3%.

## 4. Discussion

In our study of 711 patients, combination biopsies significantly outperformed targeted and additional systematic biopsies alone in the detection of csPCa (*p* < 0.001). However, there was no difference between the groups in the targeted biopsy upgrading according to the PI-RADS score (*p* = 0.651). In all PI-RADS scores, the combination biopsy was beneficial (*p* < 0.001). The gains were comparable in both the initial and repeat biopsy settings.

There are still concerns regarding the overdiagnosis of prostate cancer [14]. The U.S. Preventive Services Task Force (USPSTF) recommends against prostate biopsies based on PSA screening in 2012 [15], and they recommend only men aged 55 to 69 be measured PSA periodically in 2017. There are other predictive factors regarding prostate cancer risk, including the Prostate Health Index (PHI) and various other methods such as PSA density (PSAD) [16,17,18,19]. However, many hospitals still decide whether to conduct prostate biopsies based on the PSA level. There is little evidence that the USPSTF guidelines significantly affect people who seek prostate cancer screening [20]. Therefore, more precise biopsies and delicate patient selection are required.

When suspicious lesions are observed on mpMRI, targeted biopsies are known to be more useful than systematic biopsies [21]. However, it was somewhat unclear whether combination biopsies added additional information than targeted biopsies alone. One group argued that targeted biopsies of suspected lesions provide sufficient value for csPCa yield, and additional systematic biopsies detected mostly nonsignificant cancer [22]. Targeted biopsies do add additional information using conventional systematic biopsy methods [23], and this result is confirmed in the initial biopsy setting by a prospective multicenter trial [24]. With regard to the prostate volume, there are also reports that biopsy density is relevant in the prostate cancer detection rate [25].

Gomez-Gomez et al. reported that standard biopsy could be safely omitted in patients with an anterior lesion and in those with a PI-RADS 5 lesion. They also reported that targeted biopsy for PI-RADS 3 lesions would be less effective in detecting csPCa and thus omitted in patients with peripheral zone lesions and a previous negative biopsy [26]. In our study, though we could not find an association between G/S upgrade and biopsy methods (targeted vs. systematic), we found that regardless of the PI-RADS score and biopsy setting (initial and repeat), the detection rate of csPCa is higher in the targeted biopsy group compared with the systematic biopsy group. Moreover, our study showed the superiority of combination biopsy (targeted + additional systematic) compared with other biopsy methods.

This study demonstrated that when the mpMRI is positive (PI-RADS ≥ 3) in the initial biopsy setting, it is beneficial to combine targeted and additional systematic biopsies, as shown in the EAU guidelines [27]. According to guidelines, mpMRI is recommended prior to biopsy in patients with prior negative biopsies. Our finding suggests that combination biopsy (targeted + additional systematic) is also beneficial in a repeat biopsy setting. On the other hand, an increase from 175 to 201 cases (24.6% to 28.3%) after the addition of systematic biopsies means an increase of around 4% over 711 cases, which is still significant from a statistical point of view (*p* < 0.01) but of a debatable relevance from a clinical standpoint, given that many prediction or decision tools tend to use a 5% threshold to decide whether a certain clinical decision has to be made or not. Moreover, since the detection of insignificant cancer has also increased by about 3%, there may be a variety of considerations in terms of reducing overdiagnosis. There have been many reports regarding the usefulness of mpMRI and the PI-RADS score. However, it is necessary to note that biopsy strategy based on mpMRI has increased the cost of active surveillance [28]. With regard to cost-effectiveness, it is required to establish more specific indications for mpMRI imaging and targeted biopsies.

According to Omri et al., PSA density is associated with csPCa in radical prostatectomy specimens in small- and medium-sized prostate [29]. In our study, most patients had a small prostate size (<50 mL), PSA density was significantly associated with the PI-RADS score, and there were significant differences in G/S between the three PI-RADS groups. So, our study showed consistent results with the above study. In addition, there is a study that suggests the prostate volume obtained by MRI, either with the ellipsoid or bullet formula, proved to be almost accurate and time- and cost-effective [30]. Prostate volume was calculated based on TRUS in the above two studies, and more accurate analysis might have been possible if the prostate volume was obtained using MRI.

This study has several limitations. First, the retrospective study design performed at a single institution may have introduced inherent selection bias. Second, we did not include analysis of PI-RADS 1 and 2 lesions, because this analysis was conducted within the group of patients for whom a combination of biopsies of suspicious lesions on mpMRI was performed. Third, we could not analyze the difference according to the type of targeted biopsy (fusion vs. cognitive). There are studies that cognitive biopsy is a reliable technique, and there was no statistical significance in the difference between fusion and cognitive biopsy [31,32]. However, there is also a study that fusion biopsy is superior to cognitive biopsy [33]. Although all biopsies were performed by experienced radiologists, there might be a difference depending on the type of targeted biopsy. Fourth, our study could not suggest changes in treatment plan according to biopsy results. Additionally, we could not compare cancer lesions on mpMRI or biopsy results with radical prostatectomy specimens. There is a study that compared concordance rates regarding the side of the lesion and G/S at fusion targeted/systematic biopsy with the definitive histological report of the prostatectomy specimen [34]. Each biopsy result and whole mount histology after prostatectomy should be compared in a future study. Lastly, most importantly, the csPCa detection rate is low for our cohort. For PI-RADS 3, detection of csPCa was <5% (10/199), PI-RADS 4 was 29% (115/403), and PI-RADS 5 was 76/109 (70%). There could be some concern for generalizability. However, according to Gross et al., the cancer detection rate is lower in Asians than other races with the same PI-RADS score [35]. Our findings showed a similar rate of csPCa detection with this paper, but further study is needed on racial differences in the cancer detection rate of the PI-RADS score.

Regardless, we found that combination biopsies are advantageous when there is a suspicious lesion (PI-RADS ≥ 3) on mpMRI. The combination biopsy could be a reasonable option for csPCa detection regardless of the PI-RADS stage (3, 4, or 5). We also found that combination biopsies are beneficial in both the initial and repeat biopsy settings.

## 5. Conclusions

Combination biopsies are advantageous for the detection of csPCa in patients with a PSA level of 2.5–15 ng/mL with a suspicious lesion (PI-RADS ≥ 3) on mpMRI. Because there was no difference in the diagnosis rate change in both groups, a combination biopsy was considerable in both the initial and repeat biopsy settings.

## Figures and Tables

**Figure 1 jcm-10-04804-f001:**
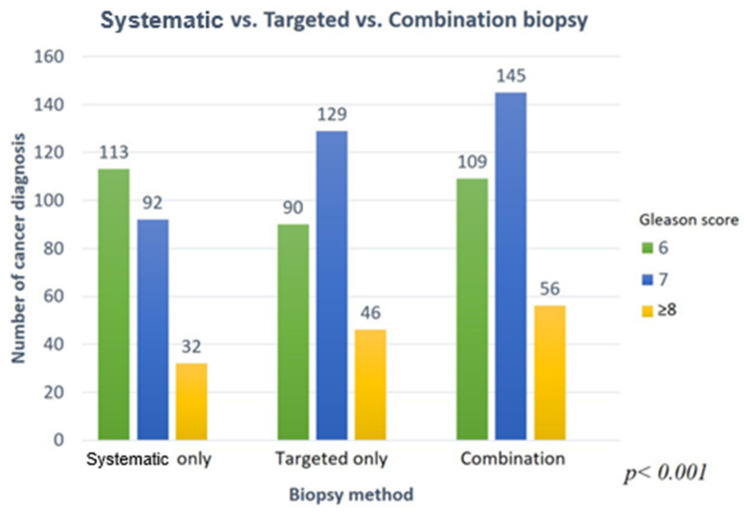
A core number of cancer diagnoses based on biopsy method.

**Figure 2 jcm-10-04804-f002:**
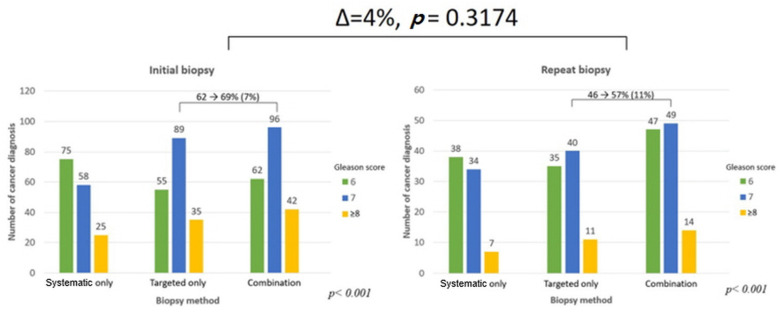
Cancer diagnosis core number in the initial and repeat biopsy settings.

**Figure 3 jcm-10-04804-f003:**
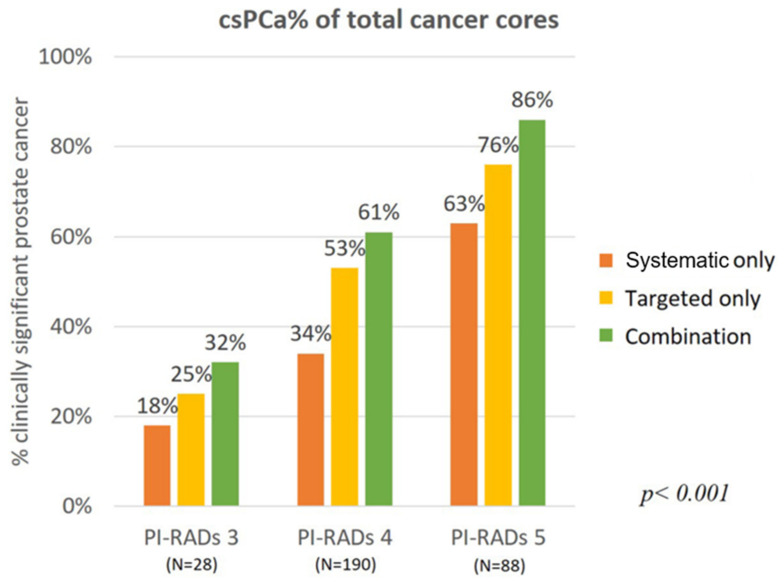
Clinically significant prostate cancer (csPCa) rate (csPCa/total cancer core) based on the PI-RADS score.

**Table 1 jcm-10-04804-t001:** Baseline demographics.

PI-RADs Score	3 (*n* = 199)	4 (*n* = 403)	5 (*n* = 109)	*p*
Age	63.0 (58.0–69.0)	64.0 (59.0–70.0)	67.0 (63.0–72.0)	<0.001
PSA (ng/mL)	4.9 (3.8–6.8)	4.8 (3.7–6.7)	6.1 (4.6–8.6)	<0.001
Biopsies (%)				0.058
Initial	102 (51.3%)	239 (59.3%)	70 (64.2%)	
Repeat	97 (48.7%)	164 (40.7%)	39 (35.8%)	
Prostate volume (mL)	42.9 (33.8–55.3)	37.3 (27.8–50.4)	31.6 (24.7–44.0)	<0.001
PSA density (ng/mL^2^)	0.14 (0.03–0.52)	0.16 (0.03–1.04)	0.23 (0.05–0.80)	<0.001
Days between biopsy and MRI (day)	51.1 (23.4–79.9)	39.5 (22.9–66.6)	31.7 (18.7–47.2)	<0.001
Total biopsy core number	12.0 (12.0–14.0)	12.0 (12.0–14.0)	12.0 (12.0–13.0)	0.069
Targeted biopsy core	3.0 (2.0–6.0)	4.0 (2.0–6.0)	3.0 (3.0–6.0)	0.017
Systematic biopsy core	10.0 (6.0–12.0)	9.0 (6.0–10.0)	8.0 (6.0–10.0)	0.032

**Table 2 jcm-10-04804-t002:** Biopsy results.

PI-RADs Group	3 (*n* = 199)	4 (*n* = 403)	5 (*n* = 109)	*p*
Cancer detection core, *n* (%)	33 (16.6%)	190 (47.1%)	88 (80.7%)	<0.001
Combination core G/S, *n* (%)				<0.001
6	23 (69.7%)	74 (39.2%)	12 (13.6%)	
7	8 (24.2%)	85 (45.0%)	52 (59.1%)	
≥8	2 (6.1%)	30 (15.9%)	24 (27.3%)	
Targeted core G/S, *n* (%)				0.003
6	14 (63.6%)	58 (36.7%)	18 (21.2%)	
7	6 (27.3%)	76 (48.1%)	47 (55.3%)	
≥8	2 (9.1%)	24 (15.2%)	20 (23.5%)	
Systematic core G/S, *n* (%)				<0.001
6	19 (79.2%)	76 (54.3%)	18 (24.7%)	
7	5 (20.8%)	47 (33.6%)	40 (54.8%)	
≥8	0 (0.0%)	17 (12.1%)	15 (20.5%)	
Targeted core upgrading (Targeted—systematic G/S)				0.651
Increased	4 (30.8%)	26 (24.1%)	18 (25.7%)	
None	9 (69.2%)	71 (65.7%)	42 (60.0%)	
Decreased	0 (0.0%)	11 (10.2%)	10 (14.3%)	

G/S, Gleason score

## Data Availability

The dataset used and/or analyzed during the current study is available from the corresponding author upon reasonable request.

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
