# Peer review of "The Role of Prostate Combination Biopsy Consisting of Targeted and Additional Systematic Biopsy"

_jcm, 2021, doi:10.3390/jcm10214804_

Round 1

Reviewer 1 Report

This study was reported the impact of combination biopsy for detecting prostate cancer. The reviewer would like to suggest some critiques as follows.

Major revision

  1. What is random biopsy? Systematic biopsy? The authors should describe the definition of random biopsy.
  2. On line 18, the authors should spell out about “PI-RADS”.
  3. On line 69, “782 patients” is wrong. “Seven hundred eighty-two patients” is correct.
  4. On line 78, the authors should quate the manuscript for evidence that PSA 2.5 ng/mL is adopted as cut-off value.
  5. On line 83, “prostatectomy pathology findings” is strange.
  6. On page 94, approximately three quarters of the enrolled patients underwent cognitive biopsy as targeted biopsy. The reviewer thinks that this is critical point in this study. The authors should the impact on results in detail.

Author Response

 Thank you for your comments.

 Your comments are very important for our manuscript.

 From now on, we provide point-by-point responses to your comments.

1. What is random biopsy? Systematic biopsy? The authors should describe the definition of random biopsy.

 - We totally agreed with your opinion. We should have described methods of biopsy clearly. Basically, systematic biopsy was performed for all patients, and the number of biopsy was determined by clinicians. We changed a word "random" to "systematic" in text as well as tables and figures, and also we mentioned abut systematic biopsy at "Materials and methods - 2.4. mpMRI and biopsy protocol" clearly. 

2. On line 18, the authors should spell out about “PI-RADS”.

 - We spelled out about "PI-RADS".

3. On line 69, “782 patients” is wrong. “Seven hundred eighty-two patients” is correct.

 - We agree with you. We changed.

4. On line 78, the authors should quate the manuscript for evidence that PSA 2.5 ng/mL is adopted as cut-off value.

 - We suggested evidence for cut-off values of PSA 2.5ng/mL at next sentence. 

5. On line 83, “prostatectomy pathology findings” is strange.

 - We clarified the meaning of the sentence by modifying existing sentence.

6. On page 94, approximately three quarters of the enrolled patients underwent cognitive biopsy as targeted biopsy. The reviewer thinks that this is critical point in this study. The authors should the impact on results in detail.

 - As your recommendation, we further described about specific target biopsy method (fusion vs. cognitive) at "Discussion - limitation" section starting with sentence "Third, we could not analyze the difference according to type of targeted biopsy (fusion vs. cognitive). "

Reviewer 2 Report

Prostate Cancer (PCa) is the second most common cancer in men, worldwide. Prostate-Specific Antigen (PSA) is a typical biomarker of prostate function. Any alteration of PSA must be framed and a prompt further investigation (e.g. biopsy) must be required. Nowadays, various methods of prostate biopsy were available (such as transrectal systematic biopsy, multiparametric MRI (mpMRI) based targeted biopsy, combination biopsy, and transperineal saturation biopsy), but there isn’t a shared consensus on which biopsy method is best for cancer detection. The aim of the current study was to compare targeted and additional random biopsy in combination biopsy methods (regarding cancer detection or detection ratio of clinically significant prostate cancer [csPCa, G/S ≥ 7]) and to evaluate which biopsy technique was better in PI-RADS 3,4 or 5 diseases based on mpMRI findings.

Comments to the Authors

Authors should be congratulated for the great contribute to the challenging topic. All future prospective should lead to improve prostate cancer detection reducing investigations number and to create new and better algorithms to properly manage early stage PCa patients, avoiding overdiagnosis and overtreatment. Despite the interesting topic, several points warrant a mention:

  1. Are data collected coming from a single center analysis? Authors should add to “limitations” section the inter-variability of biopsy interpretation between analysis centers (that represents a potential confounder).
  2. Are data available on cancer detection rate in apex PCa? It still represents a relevant concern of biopsy technique.
  3. Are data available on the following clinical decisions in these patients? Was there a clinical benefit for the patients? How this approach stretched the time between active surveillance and the surgery? Are data available on the following surgery performed? What radical prostatectomy (RP) specimens showed? Was there a concordance between biopsy and RP histological evaluation? It is worthy to discussion, this paper (doi: 10.1007/s00261-020-02798-8; PMID: 33048224) which analyzed concordance rates with definitive histologic reports between mpMRI and side of lesion and detection of clinically significant cancer (CSCD), and fusion targeted plus systematic biopsy and side of lesion and CSCD, respectively.
  4. Authors should better discuss secondary outcome such as prostate volume (PV) measurement. I suggest reading this interesting and novel paper (doi: 10.1159/000516681; PMID: 34247169) which compares multiple means of PV estimation, enlightened how PV is useful for clinical practice and patient follow-up.
  5. Authors should perform cost-benefits analysis to evaluate how the cost of active-surveillance become higher than before.

Author Response

 Thank you for your comments.

 Your comments are constructive and insightful.

 From now on, we provide point-by-point responses to your comments.

1. Are data collected coming from a single center analysis? Authors should add to “limitations” section the inter-variability of biopsy interpretation between analysis centers (that represents a potential confounder).

 - We totally agree with you, and we added the sentence to "limitation" section. 

2. Are data available on cancer detection rate in apex PCa? It still represents a relevant concern of biopsy technique.

 -  Significant number of cancers of the anterior and the apex were missed or undersampled in systematic TRUS-Bx. However, radiologists think that with significant torque, approach to these area is not impossible especially in setting of target biopsy. We added the paragraph associated with this issue in "Materials and methods - 2.4. mpMRI and biopsy protocol" section. Please see revised manuscript page 3 starting with sentence "There are concerns about risk of missing tumors of the anterior prostate and apex when performing biopsy through transrectal approach." 

3. Are data available on the following clinical decisions in these patients? Was there a clinical benefit for the patients? How this approach stretched the time between active surveillance and the surgery? Are data available on the following surgery performed? What radical prostatectomy (RP) specimens showed? Was there a concordance between biopsy and RP histological evaluation? It is worthy to discussion, this paper (doi: 10.1007/s00261-020-02798-8; PMID: 33048224) which analyzed concordance rates with definitive histologic reports between mpMRI and side of lesion and detection of clinically significant cancer (CSCD), and fusion targeted plus systematic biopsy and side of lesion and CSCD, respectively.

 - Your comment is a very important for clinicians, and this study. However, unfortunately, we could not investigated other clinical benefits specifically in active surveillance and surgery. Also, we could not deal with prostatecotmy specimens. We mentioned about these issues in "Discussion" section starting with "There is a study that compared concordance rates regarding side of lesion and G/S at fusion targeted/systematic biopsy with the definitive histologic report of prostatectomy specimen."

4. Authors should better discuss secondary outcome such as prostate volume (PV) measurement. I suggest reading this interesting and novel paper (doi: 10.1159/000516681; PMID: 34247169) which compares multiple means of PV estimation, enlightened how PV is useful for clinical practice and patient follow-up.

 - As your recommendation, we additionally discussed secondary outcome (PSA density and prostate volume) in "Discussion" section  starting with "According to Omri et al. PSA density is associated with csPCa in radical prostatectomy specimens in small and medium-sized prostate.".

5. Authors should perform cost-benefits analysis to evaluate how the cost of active-surveillance become higher than before.

 - Your opinion is essential in clinics, but unfortunately, data for active surveillance patients and costs are not available in our study. So we could not analyze them.

Round 2

Reviewer 1 Report

The authors revised the manuscript according to the reviewer’s recommendation.  The reviewer believes that this paper will provide useful information for the readers.